# Depression, anxiety, burnout and empathy among Spanish medical students

Patricia Capdevila-Gaudens[1], J. Miguel García-Abajo[2], Diego Flores-Funes[3], Mila García-Barbero[4], Joaquín García-Estañ[5]*

**1** 6th Year Student at Pompeu Fabra University, Spanish Council of Medical Students (CEEM), Barcelona, Spain, **2** 6th Year Student at the Autonomous University of Madrid, Spanish Council of Medical Students (CEEM), Madrid, Spain, **3** University Expert in Statistics and Specialist in General Surgery and Digestive System, Murcia Health Service, Murcia, Spain, **4** Former Chairwoman of the Spanish Society of Medical Education (SEDEM), University Miguel Hernandez, Alicante, Spain, **5** SEDEM and Center of Studies on Medical Education, University of Murcia, Murcia, Spain

* jgestan@um.es

**Data Availability Statement:** The data underlying the results presented in the study are available at the following repository of the Universidad de

## Abstract

Medical Education studies suggest that medical students experience mental distress in a proportion higher than in the rest of the population In the present study, we aimed to conduct a nationwide analysis of the prevalence of mental health problems among medical students. The study was carried out in 2020 in all 43 medical schools in Spain, and analyzes the prevalence of depression, anxiety, empathy and burnout among medical students (n = 5216). To measure these variables we used the Beck Depression Inventory Test for assessing depression, the Maslach Burnout Inventory Survey for Students was used for burnout, the State-Trait Anxiety Inventory (STAI) was used to assess anxiety state and trait and the Jefferson Empathy Scale 12 to obtain empathy scores. In relation to depression, the data indicate an overall prevalence of 41%, with 23.4% of participants having moderate to severe levels, and 10% experiencing suicidal ideation. Burnout prevalence was 37%, significantly higher among 6th year than among 1st year students. Anxiety levels were consistent with those reported previously among medical students (25%), and were higher than in the general population for both trait and state anxiety. The prevalence of trait anxiety was higher among women. Empathy scores were at the top end of the scale, with the highest-scoring group (>130) containing a greater percentage of women. Similarly to those published previously for other countries, these results provide a clear picture of the mental disorders affecting Spanish medical students. Medicine is an extremely demanding degree and it is important that universities and medical schools view this study as an opportunity to ensure conditions that help minimize mental health problems among their students. Some of the factors underlying these problems can be prevented by, among other things, creating an environment in which mental health is openly discussed and guidance is provided. Other factors need to be treated medically, and medical schools and universities should therefore provide support to students in need through the medical services available within their institutions.

Murcia. https://digitum.um.es/digitum/handle/10201/112325.

**Funding:** The authors received no specific funding for this work.

**Competing interests:** The authors have declared that no competing interests exist.

## Introduction

Depression is highly prevalent in our society, with an estimated 300 million people suffering from this disease [1]. In the general population, depression also affects young people and specifically medical students. Indeed, depression is the second leading cause of death among those aged between 15 and 34 years, the age range to which most medical students belong [2]. Medicine is one of the most demanding university degree courses and mental disorders are more frequent among medical students than is generally recognized. Not only depression has been frequently found in medical students. Burnout and anxiety are also important problems frequently addressed in the medical education field. Some factors that may affect the mental health of this population include high academic work load, proximity to patients' suffering, limited social life, poor family life, lack of sleep, and inconsistent and distant romantic relationships, among others. These factors can be seen as obstacles that test medical students during their training stage, setting them apart from other university students. A recent JAMA meta-analysis [1] found that 27% of 122,356 participating medical students had depressive symptoms, a higher proportion than in the general population. This exhaustive work, which analyzed a total of 168 cross-sectional and 16 longitudinal studies from 43 different countries, also provided data about suicidal ideation (11.1%). Unfortunately, these authors were unable to include any publications focusing on medical students in Spain, due to the scarcity of studies in this field.

A search in PubMed (October 6, 2020) revealed 859 articles featuring the Spanish word for "depression" and 517 featuring the Spanish term for "medical students". The union of the two terms returned only one record. The same search in EBSCO Health Sciences returned only one article that measured academic stress during the exam period. A search using the popular Google engine, however, revealed several studies not published in scientific journals; all were local analyses and reported prevalence data for poor mental health (anxiety and depression) of between 25% and 47% [3–6].

This study therefore aims to conduct a nationwide analysis of the prevalence of mental health problems among medical students. The project was approved by the Spanish Council of Medical Students (CEEM) in December 2019. As the representative of all medical students in Spain, the participation of the CEEM in the project was essential, since it ensured data were collected from a significant proportion of this population. The project, called DABE (standing for Depression, Anxiety, Burnout and Empathy, the four variables analyzed), was therefore established with the aim of determining the prevalence of these mental variables among medical students from all 43 medical schools in Spain.

Medical training in Spain follows the so-called Bologna Scheme, with 6 years of theoretical and practical training in a medical school, comprised of two basic years (1st and 2nd), a third one also basic that includes an introduction to clinical specialities, two mainly clinical years (4th and 5th) and a final 6th year, a whole clinical practical year composed of rotations in hospital services and primary care centers. In that way, medical training is usually referred as to having two cycles, a preclinical one (first three years) and a clinical cycle (last three years).

## Methods

The DABE project was a multi-center cross-sectional study. All the procedures carried out complied with the 1964 Declaration of Helsinki [7] and were approved by the Ethics Committee of the University of Murcia. The instrument used was a self-administered survey developed from a web questionnaire in Google Forms, based on a previously published questionnaire. The authors of the original questionnaire gave their explicit consent for its use [8]. The participants, all medical students, were recruited through text messages sent by the Student

Delegation Offices in each Faculty. They also gave their informed consent before completing the survey. Participation was voluntary and anonymous, and no financial remuneration was offered. The survey was active between February 17 and March 5, 2020. At that time, we estimate the total number of university undergraduates studying medicine in Spain around 42,000 subjects.

## Questionnaire

The questionnaire was divided into three parts and administered in spanish. The first part included sociodemographic and academic questions (age, year of degree course, gender, scholarship, sexual orientation, percentage of attendance at activities and part-time or full-time job). The second part comprised the DABE variables that were the object of study. To measure these variables we used the Beck Depression Inventory Test (BDI-II), a 21-question multiple-choice instrument, one of the most used psychometric scales for assessing depression [9]. Burnout was analyzed by the Maslach Burnout Inventory Survey for Students (MBI-SS) [8, 10], a tool with 15 items [8, 10]. The State-Trait Anxiety Inventory (STAI), an instrument with 20 items [11] was used to assess anxiety state (in the moment of the test or STAI-Y1) and trait (feelings habitually, STAI-Y2). Finally, the Jefferson Empathy Scale [12] was used to obtain empathy scores The scale is made up of 20 items with scoring using a 7-points Likert scale and ten of the 20 questions are valued negatively (and rectified positively in the subsequent analysis), in order to reduce the effect of acquiescence when responding. The range of possible scores goes from 20 to 140 points. The highest scores are associated with a greater degree of empathy. The third part of the questionnaire focused on potential predictors, such as academic and curricular performance and some items covering recently (previous 6 months) perceived difficulties or problems with substance abuse, skills/organization, learning, relationships, health, psychological help, social support, and adverse life events.

## Cutoff values and ranges

- Depression. For the BDI-II we used the original scoring range [9], which runs from 0 to 63. Participants are classified as having no/minimal (0–13), mild (14–19), moderate (20–28), or severe depression (29–63).

- Burnout. The three dimensions of burnout, emotional exhaustion (EX), cynicism (CY) and academic inefficiency (AI) were assessed using the MBI-SS [8, 10]. According to Galán et al. [10], the lower and upper quartiles of each burnout dimension are EX = $<1.2$ and = $>2.8$; CY = $<0.6$ and = $>2.25$; and AI = $<3.84$ and = $>5.16$. In our study, high scores on the exhaustion and cynicism subscales were considered to indicate burnout.

- Anxiety. The STAI scale [11] was used to identify individuals with high anxiety levels. It measures two types of anxiety–state anxiety, or anxiety about an event, and trait anxiety, or anxiety level as a personal characteristic. Higher scores are positively correlated with higher levels of anxiety. According to their scores, participants were classified as having very low (score 20–31), low (score 32–43), moderate (score 44–55), high (score 56–67), and very high (score 68–80) anxiety. A distinction was made between participants in the first two categories (no anxiety) and those in the latter three (suffering from anxiety), by using p75 (75th percentile).

- Empathy. It was measured using the Jefferson Empathy Scale [12], with the total empathy score being the sum of all the item scores up to a maximum possible score of 140. Higher

scores represent a more empathetic orientation. In our study, scores of over 130 were considered indicative of high empathy (percentile 80).

## Psychometric properties

All the tests used have been previously analyzed both in the general population and specifically in medical students, and found to have good psychometric properties. For instance, Galán et al [10] for Burnout, Tempski et al [13] for Anxiety, Silva et al [8] for Depression and both Ferreira-Valente et al [14] and Blanco et al [15]] for Empathy.

## Statistical analysis

First, descriptive statistics were calculated for all the variables studied, with qualitative variables being expressed in terms of absolute and relative frequency (percentages). Quantitative variables were expressed as means and standard deviations, or medians and interquartile ranges (IQR), depending on whether or not they followed a normal distribution (according to the Kolmogorov-Smirnov test). Subsequently, the inferential statistics were calculated through a series of bivariate analyses designed to test all variables in accordance with gender (female or male), year of degree (1st to 6th), Burnout Traits (Yes/No), Depression (No/Mild/Moderate/Severe), Empathy (Yes/No), Anxiety (State, Y1, Yes/No) and Anxiety (Trait, Y2, Yes/No), obtaining the Pearson' chi square. Finally, a multivariate analysis with binary logistic regression was performed for each of the main study variables, taking the main variable as the dependent variable (Burnout Yes/No, Depression Yes/No, Empathy Yes/No, Anxiety (State, Y1) Yes/No and Anxiety (Trait, Y2) Yes/No) and variables for which statistically significant results ($p < 0.05$) had been found previously in the bivariate test or those with special relevance due to their relationship with the dependent variable, as independent variables. The results were expressed in terms of the odds ratio of each independent variable, with a 95% confidence interval. Cronbach's alpha coefficient was also calculated as a means of determining internal consistency. IBM SPSS Statistics (version 24) was used for the calculations. A p-value of $<0.05$ was considered indicative of statistical significance.

## Results

A total of 5,216 students completed the survey; 76.3% were women, 22.9% men, 0.5% preferred not to say and 0.3% identified themselves as "other". The mean age (and standard deviation) of all the participants was 21.41 ± 3.44. In men, it was 21.76 ± 3.73 (IC95%, 21.6–22.0) and 21.3 ± 3.13 in women (IC95%, 21.2–21.4).

### Academic and sociodemographic data (Table 1)

The sample was drawn from all 43 medical schools in Spain, with an average of 121.3 responses per school. In relation to degree year, the highest percentage of participants were in their 1st year (20.4%), with the percentages decreasing progressively after that. As regards the percentage of engagement in non-compulsory teaching activities, over 67% of participants claimed to attend more than 50% of these classes, and 30% said they attended over 90% of them. The highest percentage of attendance was found in the 1st year of the degree course, with 43% attending more than 90% of the classes (data not shown). Most (85%) participants did not have a job; 32.7% had a scholarship of some kind and 67.3% did not, with this latter group (data not shown) having significantly more men (35.7%) than women (31.8%). Students from the 1st year of the degree course had the highest percentage of scholarships (43%), with this

**Table 1. Sociodemographic and general academic data.**

| | | Frequency | Percentage |
|---|---|---|---|
| Sex | Man | 1195 | 22,9 |
| | Woman | 3979 | 76.3 |
| | I prefer not to say | 24 | 0.5 |
| Year of study | 1st | 1063 | 20.4 |
| | 2nd | 985 | 18.9 |
| | 3rd | 929 | 17.8 |
| | 4th | 841 | 16.1 |
| | 5th | 885 | 17.0 |
| | 6th | 513 | 9.8 |
| Approximate percentage of attendance to non-compulsory classroom teaching activities | <10% | 530 | 10.2 |
| | 10–25% | 549 | 10.5 |
| | 25–50% | 593 | 11.4 |
| | 50–75% | 790 | 15.1 |
| | 75–90% | 1180 | 22.6 |
| | >90% | 1574 | 30.2 |
| Besides studying, do you work? | No | 4428 | 84.9 |
| | Yes, full time | 61 | 1.2 |
| | Yes part time | 727 | 13.9 |
| Do you have a scholarship of any kind? | No | 3511 | 67.3 |
| | Yes | 1705 | 32.7 |
| Sexual orientation | Heterosexual | 4029 | 77.2 |
| | Homosexual | 254 | 4.9 |
| | Bisexual | 737 | 14.1 |
| | I prefer not to say | 196 | 3.8 |

figure being around 30% in subsequent years (data not shown). In relation to sexual orientation, 77% identified themselves as heterosexual, 14.1% as bisexual, 4.9% as homosexual and 3.8% preferred not to answer. The percentage of those identifying themselves as homosexual was significantly higher in men (14.6%) than in women (1.9%), whereas the percentage of women who identified themselves as bisexual was significantly higher (14.6% as compared to 11.4% among men). These last data are not shown.

## Perceived mental health (Table 2)

1. Depression: The mean value was 13.46 ± 10.61, with a 95% CI from 13.17 to 13.75. A total of 41% of students reported some symptoms of depression and the percentage of women was significantly higher (43.1%) than that of men (33.6%). Moreover, 17.6% were found to have mild depression, 13.2% moderate depression and 10.2% severe depression, with all percentages being higher among women. In relation to question 9 of Beck's test (suicidal ideation), 89% claimed not to have had ideas about harming themselves, although the rest (11%) did acknowledge suicidal ideation to varying degrees.

2. Anxiety: 24.7% of students had high anxiety at the time of the survey (state), while 21.5% had high levels of anxiety on a regular basis (trait). The percentage of women with trait anxiety was significantly higher than that of men. No significant differences were observed between students in different years of their degree course.

**Table 2. Percentages of DABE variables in men and women.**

| | Men | | | |
|---|---|---|---|---|
| **DEPRESSION** | **No/Minimal** | **Mild** | **Moderate** | **Severe** |
| 1st | 72,07 % | 13,06 % | 8,11 % | 6,76 % |
| 2nd | 68,32 % | 13,86 % | 10,89 % | 6,93 % |
| 3rd | 63,72 % | 15,81 % | 10,70 % | 9,77 % |
| 4th | 60,48 % | 16,67 % | 12,38 % | 10,48 % |
| 5th | 63,59 % | 18,89 % | 9,22 % | 8,29 % |
| 6th | 72,09 % | 10,85 % | 8,53 % | 8,53 % |
| **Mean** | **66,36 %** | **15,15 %** | **10,04 %** | **8,45 %** |
| | Women | | | |
| **DEPRESSION** | **No/Minimal** | **Mild** | **Moderate** | **Severe** |
| 1st | 63,23 % | 15,57 % | 12,46 % | 8,74 % |
| 2nd | 53,85 % | 18,46 % | 14,23 % | 13,46 % |
| 3rd | 51,84 % | 19,12 % | 17,28 % | 11,76 % |
| 4th | 55,04 % | 17,76 % | 16,64 % | 10,56 % |
| 5th | 58,75 % | 19,48 % | 11,87 % | 9,89 % |
| 6th | 58,78 % | 21,81 % | 11,97 % | 7,45 % |
| **Mean** | **56,92 %** | **18,35 %** | **14,17 %** | **10,56 %** |
| **ANXIETY state** | **All** | **Men** | **Women** | |
| 1st | 26,90 % | 27,48 % | 26,71 % | |
| 2nd | 25,79 % | 27,72 % | 25,26 % | |
| 3rd | 25,62 % | 20,00 % | 26,91 % | |
| 4th | 22,71 % | 23,33 % | 22,40 % | |
| 5th | 21,92 % | 19,35 % | 22,83 % | |
| 6th | 24,56 % | 23,26 % | 25,27 % | |
| **Mean** | **24,71 %** | **23,51 %** | **25,01 %** | |
| **ANXIETY trait** | **All** | **Men** | **Women** | |
| 1st | 21,54 % | 18,47 % | 22,51 % | |
| 2nd | 22,03 % | 18,81 % | 22,82 % | |
| 3rd | 23,90 % | 19,53 % | 24,93 % | |
| 4th | 20,21 % | 16,19 % | 21,76 % | |
| 5th | 21,02 % | 16,59 % | 22,37 % | |
| 6th | 19,30 % | 13,18 % | 21,28 % | |
| **Mean** | **21,53 %** | **17,41 %** | **22,74 %** | |
| **BURNOUT** | **All** | **Men** | **Women** | |
| 1st | 22,58 % | 22,52 % | 22,40 % | |
| 2nd | 35,23 % | 29,70 % | 36,67 % | |
| 3rd | 38,21 % | 34,42 % | 39,38 % | |
| 4th | 39,60 % | 36,67 % | 40,48 % | |
| 5th | 46,67 % | 48,85 % | 46,12 % | |
| 6th | 44,83 % | 40,31 % | 46,28 % | |
| **Mean** | **36,77 %** | **35,06 %** | **37,22 %** | |
| **EMPATHY** | **All** | **Men** | **Women** | |
| 1st | 14,02 % | 13,51 % | 14,13 % | |
| 2nd | 15,33 % | 7,92 % | 17,31 % | |
| 3rd | 20,02 % | 11,16 % | 22,80 % | |
| 4th | 19,98 % | 12,86 % | 22,56 % | |
| 5th | 22,03 % | 14,29 % | 24,51 % | |

*(Continued)*

**Table 2.** (Continued)

| | | | | |
|---|---|---|---|---|
| 6th | 26,12 % | 23,26 % | 27,66 % | |
| **Mean** | **18,85 %** | **13,22 %** | **20,61 %** | |

3. Burnout: 36.8% of participants had high burnout, defined as high scores for two of its components (exhaustion and cynicism). No differences were observed between men and women. The percentage of high burnout increased progressively from 1st (23%) to 6th (45%) year.

4. Empathy: The mean numerical value for this variable was 120.6 ± 11.8 points (maximum 140), with 18.8% of participants having high levels of empathy (>130). The percentage of women was significantly higher than that of men, and percentages increased progressively year by year (from 14% in the 1st to 26% in the 6th year of the course).

## Bivariant analysis between mental variables (Table 3)

Depression was significantly related to anxiety trait, burnout and empathy, so that individuals having signs of depression had greater levels of anxiety trait and burnout but lower empathy levels. Having anxiety state was related only to high anxiety trait. High anxiety trait was significantly associated with depression, anxiety state and burnout, thus individuals with high anxiety trait had greater levels of these three variables. Burnout was significantly associated with both depression and anxiety trait, whereas empathy was significantly related only to depression, so that high empathic individuals showed lower depression signs.

## Potential risk predictors (Table 4)

The main problem or difficulty perceived by medical students was how to organize academic work (time management), with the percentage of women being significantly higher than that of men (77% vs. 70%). Academic performance problems varied across the years of the degree course, with peaks among students in their 2nd, 3rd and 4th years. In relation to events that had occurred over the past 6 months, 25% of participants mentioned someone important to them having had a serious illness or accident, with no differences observed in this sense between men and women. Moreover, 21.7% mentioned financial problems, 17% the end of a stable

**Table 3. Association between variables.**

| | | DEPRESSION | | | HIGH ANXIETY STATE | | | HIGH ANXIETY TRAIT | | | HIGH BURNOUT | | | HIGH EMPATHY | | |
|---|---|---|---|---|---|---|---|---|---|---|---|---|---|---|---|---|
| | | NO | YES | Chi$^2$ p level | NO | YES | Chi$^2$ p level | NO | YES | Chi$^2$ p level | NO | YES | Chi$^2$ p level | NO | YES | Chi$^2$ p level |
| **Depression** | Yes | - | - | - | 1623– 41.3% | 518– 40.2% | 0.469 | 1514– 37.0% | 627– 55.8% | 0.000 | 809– 24.5% | 1332– 69.4% | 0.000 | 1783– 42.1% | 358– 36.4% | 0.001 |
| **Anxiety (State)** | High | 771– 25.1% | 518– 24.2% | 0.469 | | | | 745– 18.2% | 544– 48.4% | 0.000 | 824– 25.0% | 465– 24.2% | 0.55 | 1047– 24.7% | 242– 24.6% | 0.94 |
| **Anxiety (Trait)** | High | 496– 16.1% | 627– 29.3% | 0.000 | 579– 14.7% | 544– 42.2% | 0.000 | | | | 614– 18.6% | 509– 26,5% | 0.000 | 910– 21.5% | 213– 21.7% | 0.907 |
| **Burnout** | High | 586– 19.1% | 1332– 62.2% | 0.000 | 1453– 37.0% | 465– 36.1% | 0.55 | 1409– 34.4% | 509– 45.3% | 0.000 | | | | 1561– 36.9% | 357– 36.3% | 0.743 |
| **Empathy** | Yes | 625– 20.3% | 358– 16.7% | 0.001 | 741– 18.9% | 242– 18.8% | 0.94 | 770– 18.8% | 213– 19.0% | .907 | 626– 19.0% | 357– 18.6% | 0.743 | | | |

Data are number and percentage.

**Table 4. Predictive factors.**

| | | Frequency | Percentage |
|---|---|---|---|
| **Problems or difficulties encountered** | **Academic time management** | 3938 | 75.5% |
| | **Academic performance** | 3250 | 62.3% |
| | **Physical health** | 2935 | 56.3% |
| | **Task organization** | 2535 | 48.6% |
| | **Money management** | 1759 | 33.7% |
| | **Family relationships** | 1596 | 30.6% |
| | **Colleagues relationships** | 1533 | 29.4% |
| | **Partner relationships** | 1207 | 23.1% |
| | **Death of a family member** | 846 | 16.2% |
| | **Tobacco abuse** | 361 | 6.9% |
| | **Teachers relationship** | 304 | 5.8% |
| | **Alcohol abuse** | 242 | 4.6% |
| **List of events** | **Serious illness or accident in someone close to me** | 1309 | 25.1% |
| | **Economic problems** | 1131 | 21.7% |
| | **End of a stable love relationship** | 896 | 17.2% |
| | **Serious illness or accident of mine** | 506 | 9.7% |
| | **Abuse of other drugs** | 100 | 1.9% |
| **Social support 1. How many people are you close enough to if you experience serious personal problems?** | **>5** | 2122 | 40.7% |
| | **3–5** | 2079 | 39.9% |
| | **1 o 2** | 948 | 18.2% |
| | **None** | 67 | 1.3% |
| **Social support 2. What is your interest and participation in your everyday experiences?** | **Much** | 1963 | 37.6% |
| | **Something** | 1849 | 35.4% |
| | **I don't know** | 868 | 16.6% |
| | **Little** | 483 | 9.3% |
| | **None** | 53 | 1.0% |
| **Social support 3. How satisfied are you with the social activities you currently participate in?** | **Much** | 1815 | 34.8% |
| | **Something** | 1878 | 36.0% |
| | **I don't know** | 483 | 9.3% |
| | **Little** | 858 | 16.4% |
| | **None** | 182 | 3.5% |
| **Social support 4, At this moment, how satisfied are you with the support you receive from your social relationships?** | **Much** | 2407 | 46.1% |
| | **Something** | 1575 | 30.2% |
| | **I don't know** | 493 | 9.5% |
| | **Little** | 601 | 11.5% |
| | **None** | 140 | 2.7% |
| **Academic performance 1. Academic classification in relation to the effort made in the course** | **Lower than the effort made** | 2528 | 48.5% |
| | **Concordant effort** | 2390 | 45.8% |
| | **Greater than the effort made** | 298 | 5.7% |
| **Academic performance 2. Satisfaction with academic grades at this time** | **Very much satisfied** | 493 | 9.5% |
| | **Satisfied** | 2214 | 42.4% |
| | **Little satisfied** | 1810 | 34.7% |
| | **No satisfied** | 699 | 13.4% |

(*Continued*)

**Table 4.** (Continued)

| | | Frequency | Percentage |
|---|---|---|---|
| **Academic performance 3. Level of satisfaction of your parents with your academic performance** | **Very much satisfied** | 2422 | 46.4% |
| | **Satisfied** | 1915 | 36.7% |
| | **Little satisfied** | 638 | 12.2% |
| | **No satisfied** | 241 | 4.6% |
| **Tobacco smoker** | **Yes** | 407 | 7.8% |
| | **No** | 4392 | 84.2% |
| | **Ocassional** | 417 | 8.0% |
| **Cannabis smoker** | **No, never** | 3135 | 60.1% |
| | **I have tried** | 1782 | 34.2% |
| | **Ocassionally, < = 1 per week** | 227 | 4.4% |
| | **Habitually, 2 or more times per week** | 68 | 1.3% |
| **Alcohol** | **Never** | 829 | 15.9% |
| | **1 or less a month** | 1762 | 33.8% |
| | **2–4 a month** | 1984 | 38.0% |
| | **2 a week** | 526 | 10.1% |
| | **More than 3 a week** | 115 | 2.2% |
| **Psychopharmaceuticals** | **Yes** | 1063 | 20.4% |
| | **No** | 4153 | 79.6% |
| | **Tranquilizers / Anxiolytics** | 851 | 16.3% |
| | **Mood stabilizers** | 28 | 0.5% |
| | **Antidepressants** | 532 | 10.2% |
| | **Neuroleptics / antipsychotics** | 57 | 1.1% |
| | **Other** | 121 | 2.3% |

romantic relationship and 16% the death of a family member. Just under 10% mentioned having had a serious personal illness or accident (9.7%).

In relation to social support, almost 80% said they had more than 3 people they could count on if they had serious personal problems, with no differences being observed between men and women. The percentage of students with more than 5 people they could count on increased the further along on their degree course they were.

In terms of interest and participation in daily experiences, almost 10% claimed to have little or no interest, with this percentage being significantly lower among women. One fifth (20%) reported gaining little or no satisfaction from the social activities in which they participated and 13% reported receiving little or no support from their social relationships, with no differences being observed between men and women in these last two cases.

As for academic performance, almost 50% of students claimed to have grades that were lower than they had expected given the effort made, and considered themselves to be little or not at all satisfied with their current academic performance.

## Drugs and substance abuse (Table 4)

Of the sample, 16% said they smoked, with this percentage being higher among men (20%) than among women (14%), and increasing as students progressed through their degree. As regards cannabis, 60% claimed never to have tried it and 1.3% said they used it habitually, with this percentage being higher among men than in women and also increasing over time (data not shown). Just over a tenth of all participants (12%) reported consuming alcohol two or more times per week, with this percentage again being higher among men (18%) than among

Table 5.  Risk factors for depression.

| DEPRESSION | P value | Odds Ratio | I.C. 95% for Odds Ratio | |
|---|---|---|---|---|
| | | | Lower | Upper |
| Low interest and participation in daily activities | <0.001 | 5.115 | 3.836 | 6.819 |
| High Burnout | <0.001 | 3.739 | 3.207 | 4.359 |
| Physical health problems | <0.001 | 2.673 | 2.292 | 3.116 |
| Low satisfaction with academic grades | <0.001 | 2.208 | 1.896 | 2.571 |
| Take psychotropic drugs | <0.001 | 2.185 | 1.819 | 2.624 |
| Problems of relationship with colleagues | <0.001 | 2.097 | 1.785 | 2.463 |
| Academic performance problems | <0.001 | 1.994 | 1.683 | 2.362 |
| Problems of relationship with family | <0.001 | 1.921 | 1.637 | 2.253 |
| High Anxiety (Trait) | <0.001 | 1.780 | 1.472 | 2.153 |
| Problems of relationship with partner | <0.001 | 1.767 | 1.485 | 2.102 |
| Being woman | <0.001 | 1.561 | 1.295 | 1.882 |
| Academic homework organization problems | <0.001 | 1.469 | 1.263 | 1.708 |
| Homo-bisexual orientation | <0.001 | 1.442 | 1.212 | 1.716 |
| Habitual tobacco use | 0.011 | 1.300 | 1.062 | 1.592 |
| Academic work organization problems | 0.024 | 1.241 | 1.029 | 1.496 |
| Academic homework organization problems | <0.001 | 1.469 | 1.263 | 1.708 |
| 4th-6th year | 0.004 | .797 | .684 | .928 |

women (10%). One fifth (20%) of participants reported taking psychotropic drugs, mainly anxiolytics (16.3%) and antidepressants (10.2%). Women took more psychotropic drugs (21%) than men (16%) and those in later years of the degree course took more than their counterparts in earlier ones, with the percentage increasing from 13% in the 1st to 23% in the 6th year of the course in relation to anxiolytics, and from 7% to 12% in relation to antidepressants.

## Multivariable relationships between DABE variables and risk predictors

Tables 5 to 8 show the strength of association among the DABE variables and between them and the risk factors studied. Only factors found to be significant are included. In the case of depression (Table 5), low interest and participation in daily activities were the main risk factors

Table 6.  Risk factors for burnout.

| BURNOUT | P value | Odds Ratio | I.C. 95% for Odds Ratio | |
|---|---|---|---|---|
| | | | Lower | Upper |
| Depression symptoms | <0.001 | 3.770 | 3.242 | 4.385 |
| Problems of academic performance | <0.001 | 2.163 | 1.851 | 2.528 |
| 4th-6th year | <0.001 | 1.895 | 1.653 | 2.173 |
| Academic satisfacción lower to effort | <0.001 | 1.713 | 1.496 | 1.961 |
| Low satisfaction with social activities | <0.001 | 1.667 | 1.411 | 1.969 |
| Taking psychopharmaceuticals | <0.001 | 1.431 | 1.217 | 1.682 |
| Academic homework organization problems | <0.001 | 1.387 | 1.167 | 1.649 |
| Physical health problems | 0.001 | 1.292 | 1.117 | 1.494 |
| Homo-bisexual orientation | 0.004 | 1.259 | 1.078 | 1.469 |
| High Anxiety (Trait) | 0.034 | 1.206 | 1.014 | 1.435 |
| Problems of relationship with family | 0.026 | 1.181 | 1.020 | 1.366 |
| Assistance to no compulsory activities >50% | 0.001 | 0.785 | 0.682 | 0.904 |

**Table 7. Risk factors for anxiety.**

| ANXIETY-STATE | P value | Odds Ratio | I.C. 95% for Odds Ratio | |
|---|---|---|---|---|
| | | | Lower | Upper |
| High Anxiety Trait | <0.001 | 6.621 | 5.322 | 8.238 |
| Work in addition to studying | 0.002 | 1.540 | 1.166 | 2.033 |
| Money management problems | 0.026 | 1.290 | 1.031 | 1.614 |
| 4th-6th year | 0.004 | 0.721 | 0.575 | 0.903 |
| Low social support 3 | <0.001 | 0.551 | 0.404 | 0.752 |
| ANXIETY-TRAIT | P value | Odds Ratio | I.C. 95% for Odds Ratio | |
| | | | Lower | Upper |
| High Anxiety State | <0.001 | 6.748 | 5.395 | 8.441 |
| Depression symptoms | <0.001 | 1.737 | 1.432 | 2.106 |

(associated with a five-fold risk of having depressive symptoms, with a range of 3.8 to 6.8), followed by high burnout and physical health problems, to cite only the first three. The odds ratio for burnout (Table 6) again revealed an association with depression (odds ratio of 4), with academic performance problems and engaging in clinical rotations being the following two most important factors. The odds ratio values for anxiety (Table 7) indicated that both types, trait and state, were closely related to each other, as well as to working while studying and having financial problems. Finally, as shown in Table 8, empathy was closely associated with being a woman, engaging in clinical rotations and having a high level of social support.

## Discussion

In this study, which was carried out in 2020 in all 43 medical schools in Spain, we analyze the prevalence of depression, anxiety, empathy and burnout among medical students, in the first nationwide analysis of these mental health variables in this particular population. Although previous studies in this field have reported findings from some isolated university centers [3–6], ours is the first to offer global data from across the entire country. Moreover, the sample (5,216 participants) is large enough to endow the data presented here with a high degree of reliability, thereby enabling conclusions to be drawn. At present, the total number of university undergraduates studying medicine in Spain is close to 42,000, since around 7,000 freshmen began their studies in September 2019 [16]. This means that our sample represents around 12% of the total number of medical students in Spain. It is also important to highlight the fact that the survey, which was carried out at the beginning of the second semester, includes responses from students studying at all 43 medical schools operating at that time in Spain. We therefore believe that the number of participants may reflect the true situation of the problem in Spain medical training.

In relation to depression symptoms, the data indicate an overall prevalence of 41%, with the figure for moderate and severe depression being 23.4%. These findings are similar to those reported in the local studies cited above, being slightly lower than the figure reported for 4th

**Table 8. Risk factors for empathy.**

| EMPATHY | P value | Odds Ratio | I.C. 95% for Odds Ratio | |
|---|---|---|---|---|
| | | | Lower | Upper |
| Being woman | <0.001 | 1.726 | 1.435 | 2.077 |
| 4th-6th year | <0.001 | 1.503 | 1.306 | 1.730 |
| High social support 3 | <0.001 | 1.430 | 1.106 | 1.847 |

year students at Catalonian medical schools (47%) [3] and practically identical to those found at the Universities of Murcia and Valencia [4–6]. In all these studies, as in the present work, depression was found to be more prevalent among women than among men. This is consistent also with the data reported in practically all international studies carried out on this topic [1]. Our results for depression symptoms are also consistent with those published recently in relation to some Italian medical schools, as well as with the findings of systematic reviews in both Europe and worldwide [17–19]. We can therefore conclude that Spanish medical students behave similarly to their counterparts in other countries, indicating a certain degree of constancy in terms of problems or inductive factors throughout university centers worldwide. Our findings are also consistent with previous research reporting that the prevalence of depression symptoms is higher among medical students than in the general population, and also higher among women. For instance, in Spain, according to data from the 2017 National Health Survey [20], depression is more than twice as prevalent among women in the general population (9.2%) than among men (4%). In relation to the prevalence of depression symptoms among university students from disciplines other than medicine, the data are inconclusive [21], with percentages being either similar or lower [22]. Two of the most important factors associated with depression in our study were health issues and dissatisfaction with academic performance, which may well be the cause of the problem, while others such as lack of interest and participation in daily activities and consumption of psychotropic drugs may be its consequences.

One important finding in relation to the study of depression is the percentage of suicidal ideation found among Spanish medical students, in relation to which our results are consistent with those published previously [1, 3–6]. As stated earlier, around 10% of medical students admit to having engaged in suicidal ideation, a figure that indicates an urgent need to establish preventive measures and improve students' access to advice, care and treatment to help them overcome this problem. Unfortunately, there are around 3,500 suicides every year in Spain (3,539 in 2018), of which 7% are young people between the ages of 15 and 29 [23]. However, no data is available on specific suicide rates among university students [24], although higher suicide rates have been reported among physicians than among the general population (1.3% as opposed to 0.8%) [25].

Regarding burnout, our data are consistent with those found among 3rd and 6th year medical students from a Spanish university [6], among which burnout risk was significantly more prevalent among 6th year students (37.5%) than among those in their 3rd year of training (14.8%), with no association being found between burnout and gender. In our study, burnout scores increased after the 1st year of the medical degree, with the highest scores being found in the 5th year. However, the means were very similar (around 35%), suggesting that the prevalence of burnout among medical students is fairly similar to that found among medical residents and practicing physicians [26, 27]. We identified several factors that may contribute to burnout among medical students, including curricular factors, personal life events, and the learning environment [14, 28, 29]; and burnout in turn may have serious consequences, leading to unprofessional conduct, increased risk of suicidal ideation, and serious thoughts of dropping out of medical school. However, some authors have suggested that burnout among medical students may be reversible, with 26% of sufferers recovering within a year [28]. We therefore believe it is vital to identify and treat students suffering from this syndrome before they begin their medical residencies.

In relation to empathy, the scores obtained by our sample were at the upper end of the scale [12, 15] and the mean value observed is consistent with that reported by previous studies both in Spain and abroad [15, 30–33], and even slightly higher than in some. As stated in the Results section, women scored higher than men, as did those engaging in clinical rotations and those

with strong social support, as indeed has been reported previously by other authors [34, 35]. It therefore appears that empathy levels increase over time and specially when students enter into contact with patients during their clinical rotations [36, 37].

The anxiety levels found in our study are consistent with the values reported previously [7, 11] among medical students, and are higher than in the general population in both categories or scales (trait and state). Furthermore, the greater prevalence of trait anxiety among women observed in our results has been also reported previously [11, 38, 39]. In relation to risk factors, the two categories were closely related to each other, and state anxiety particularly was very frequent among students who worked while studying, as well as among those experiencing financial difficulties. Finally, although no statistical differences were observed between students on different years of the degree course, those engaging in clinical rotations did seem to be somewhat protected against state anxiety, probably because they were better able to cope with medical school activities.

## Strengths and limitations

The nationwide nature of the study design is a factor that speaks to the generalizability of our results; however, although the number of responses was high, not all the results will be applicable to each and every university, since the response rate varied widely across centers (from 0.2% to 5.8% of the total number of responses). The use of a self-administered instrument rather than structured clinical interviews may also be a limitation, since although used in many studies, depression scores may not accurately reflect the severity of depression. However, since it was not possible to perform a clinical evaluation of all medical students in Spain, we believe that the approach adopted provides a clear picture of something that is often commented on by students themselves, and which may lead to stigmatization [40]. It is also possible that students who were especially sensitive to these issues or were aware of having these problems were more predisposed to complete the survey than others. The moment at which surveys are launched may also be a confounding factor. We therefore decided to start the survey at the beginning of the second semester (which usually commences during the last week of January in most Spanish universities), once all the first semester examinations had finished and students had no responsibilities other than attending theoretical classes or practical and hospital rotations. Also, we only captured individual constructs, whereas institutional constructs that may also contribute were not analysed. Finally, it is important to emphasize that the instruments used in the present study are not diagnostic but of a screening nature. Finally, the sampling procedure was non probabilistic, since participation was offered to everyone but only those who wanted participated.

## Conclusions

In the present study, we report the prevalence of depression, anxiety, burnout and empathy among students from all the medical schools in Spain. The results, which are similar to those already published in relation to other countries, provide a clear picture of the mental disorders affecting Spanish medical students. Medicine is an extremely demanding degree and it is important that universities and medical schools view this study as an opportunity to ensure conditions that help minimize mental health problems among their students. Some of the factors underlying these problems can be prevented by, among other things, creating an environment in which mental health is openly discussed and guidance is provided. Other factors need to be treated medically, and medical schools and universities should therefore provide support to students in need through the medical services available within their institutions.

## Acknowledgments

The authors wish to thank the following students for helping them with the survey: Sònia Serrabou Pradas, Universidad Autónoma de Barcelona; Jorge Monzón Moya, Universidad de Alcalá de Henares; Lucía Martín López, Universidad Autónoma de Madrid; Ignacio Coll Orduña; Universidad Alfonso X El Sabio; Alba Arbués Martí, Universidad de Barcelona- Bellvitge; Mar Niella Bellot, Universidad de Barcelona-Clinic; María Sasía, Universidad de Cantabria; Ángel Francisco Serrano Andrades, Universidad de Cádiz; Ander Santiago Gómez, Universidad CEU Cardenal Herrera–Castelló; Sara Ivars Bernal, Universidad CEU Cardenal Herrera- Valencia; Laura Sindín, Universidad San Pablo C.E.U.; Juan Fco Vázquez Rodríguez, Universidad de Castilla La Mancha-Ciudad Real; Beatriz Jarabo Tévar, Universidad de Castilla La Mancha-Albacete; Jorge Mesa Guadalupe, Universidad Complutense de Madrid; Jesús Canales Fernández, Universidad de Córdoba; Alba María Torrat Novés, Universidad Católica de Valencia S. Vicente M.; Maria Molina Guerrero, Universidad de Girona; Alysson Micolta, Universidad Europea de Madrid; Clara Hoyas Sánchez, Universidad de Extremadura; Daniela Rocca Flores, Universidad Francisco de Vitoria; Araceli Jiménez Lara, Universidad de Granada; Yeray Damas Pons, Universitat de las Illes Balears; Marc Vallés Alabart, Universidad Internacional de Cataluña; Chourok Aknin, Universitat Jaume I de Castellón; Iriana Rodríguez Hernández, Universidad La Laguna; Rosa Maria Navarro Lopez, Universidad de Murcia; Yosef Benzaquen Aserraf, Universidad de Málaga; Laura Sánchez Sánchez, Universidad Miguel Hernández de Elche; Alberto Lorenzo Longedo, Universidad de Oviedo; Patricia Capdevila, Universitat Pompeu Fabra; Rodrigo Guitian Montes, Universidad del País Vasco; Lucas Antigono, Universidad Rey Juan Carlos; Bruno Fernández Cuevas, Universidad de Sevilla; Gloria Corbacho Hernández, Universidad de Salamanca; Candela Fernández Reino, Universidad de Santiago de Compostela; Jesús Andicoberry López, Universitat de València; Mateo Jiménez García, Universidad de Valladolid; Ana García Blasco, Universidad de Zaragoza; Mar García Bellas, Universidad las Palmas de Gran Canaria; Jose Carte García, Universidad de Navarra.

We also thank Diana Draper for correcting the english manuscript.

## Author Contributions

**Conceptualization:** Patricia Capdevila-Gaudens, J. Miguel García-Abajo, Mila García-Barbero, Joaquín García-Estañ.

**Data curation:** Diego Flores-Funes, Joaquín García-Estañ.

**Formal analysis:** Joaquín García-Estañ.

**Investigation:** J. Miguel García-Abajo, Diego Flores-Funes, Joaquín García-Estañ.

**Methodology:** Patricia Capdevila-Gaudens, Diego Flores-Funes, Joaquín García-Estañ.

**Project administration:** Patricia Capdevila-Gaudens, Mila García-Barbero, Joaquín García-Estañ.

**Resources:** J. Miguel García-Abajo, Diego Flores-Funes.

**Software:** Diego Flores-Funes.

**Supervision:** Patricia Capdevila-Gaudens, Mila García-Barbero, Joaquín García-Estañ.

**Validation:** Mila García-Barbero.

**Writing – review & editing:** Patricia Capdevila-Gaudens, J. Miguel García-Abajo, Diego Flores-Funes, Mila García-Barbero, Joaquín García-Estañ.

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
