## [Decision Letter · Decision Letter 0]

5 Aug 2021

PONE-D-21-11578

Depression, Anxiety, Burnout and Empathy among Spanish Medical Students

PLOS ONE

Dear Dr. Garcia-Estañ,

Thank you for submitting your manuscript to PLOS ONE. After careful consideration, we feel that it has merit but does not fully meet PLOS ONE’s publication criteria as it currently stands. Therefore, we invite you to submit a revised version of the manuscript that addresses the points raised during the review process.

We look forward to receiving your revised manuscript.

Kind regards,

Kamran Sattar

Academic Editor

PLOS ONE

Journal Requirements:

Reviewers' comments:

Reviewer's Responses to Questions

**Comments to the Author**

1. Is the manuscript technically sound, and do the data support the conclusions?

Reviewer #1: Partly

2. Has the statistical analysis been performed appropriately and rigorously? 

Reviewer #1: No

3. Have the authors made all data underlying the findings in their manuscript fully available?

Reviewer #1: Yes

4. Is the manuscript presented in an intelligible fashion and written in standard English?

Reviewer #1: No

5. Review Comments to the Author

Reviewer #1: The manuscript offers an overview on medical students mental health in Spain. The strength of the study was the national scale study. However, I feel that the manuscript need to be improved based on the following comments before it is ready for publication.

This manuscript would need a native speaker proofreading to increase the readability of the findings. There is a glaring grammatical and spelling (eg: PubMed) errors throughout the manuscript.

Abstract:

Need to include study objective and some general methodology before jumping to results. Worthwhile to include the tools in the abstract so that readers can grasp whether the depression/anxiety prevalence was more of positive symptoms or diagnosis.

Introduction:

1. The introduction focused only on depression while the constructs measured also include burnout, anxiety and empathy. Hence, authors could incorporate this flow in the introduction

Introduction on burnout, depression and anxiety and some overview in general population - High prevalence of burnout, depression and anxiety - why it is more pronounced among medical students - the impact of these mental health problems if not addressed (including empathy) - why empathy is important in medical training - lack of studies looking into this in Spanish context

2. A search in Pubmed (October 6, 2020) revealed 859 articles featuring the Spanish word for "depression" and 517 featuring the Spanish term for “medical students". The union of the two terms returned no records at all.

(From my search, it seems there is such study (PMID: 32920780). Hence, this paragraph should be updated.

3. Some introduction on medical training and the structure in Spain is necessary since this is a national scale study. It would help readers to understand which are pre and clinical year students.

Methods:

1. First para: Need to include population size or estimate to give some picture to readers.

2. Each instruments need to be introduced separately in individual paragraph for a better flow- purpose of the instrument (screening or diagnosis), whether it was administered in English or Spanish version, validation in Spanish or medical students context, with reliability mentioned - scores calculation (whether there is dichotomous classification or measured as continuous variable). Note that validation must be mentioned specifically for the instruments (eg. CBI) rather than the construct (eg. Burnout)

3. Some explanation is needed to help readers in the difference of anxiety state vs trait

4. What is p75?

5. Statistical analysis - more details are needed what are the independent variables included for each logistic regression analyses? How was the independent variables selected for the multivariable model? (Some study used cutoff of p-value more than 0.2 for each univariate analysis run)

Results

1. "Cronbach's alpha coefficients were 0.923 (Beck's test), 0.703 (Maslach), 0.809 (Jefferson), 0.378 (state anxiety) and 0.365 (trait anxiety)." - I think this is not necessary as validation should be done in a separate cohort rather than the actual sample.

2. Table 1 - Percentage (%) should be in form of 22.9 rather than 22,9.

3. Table 1 - Age should be excluded and just mentioned in the paragraph in form of mean and standard deviation.

4. Second para: Authors refer this para as Table 1 but it seems there were 6 subgroup analyses that were not in Table 1. These subgroup analyses were mentioned selectively rather than comprehensively. Hence, authors should rethink whether this subgroup analysis need a separate table.

5. I suggest that Figure 1 to Figure 5 is changed into Table. It is difficult to read the result precisely from the bar chart.

6. "Percentage of depression was also significantly higher among 2nd, 3rd, and 4th year students, with those in their 1st and 6th years scoring lowest in this sense (Figure 2)" - How did authors came to this statement. Was one-way ANOVA conducted to ascertain this? - This comment also applies to paragraph on anxiety, burnout and empathy.

7. There was untranslated Spanish words in Table 3.

8. Drugs and substance abuse: "... with this percentage again being significantly higher among men.." - Authors must be careful when saying significantly unless there was t test or anova done to ascertain the association.

9. Multivariate relationships between DABE variables and risk predictors - Multivariable instead of multivariate

10. Table 4-7 (Inferior superior should be named as lower and upper)

11. p value of 0.000 should be written as p-value < 0.001

Discussion

1. "We therefore believe that the results offer an accurate and highly relevant overview of the mental health of medical students in our country" - I think this is overinflation of the study, plus it is a non-probability sampling. A better way to say this is - the number of participants may reflect the true situation of the problem in Spain medical training.

2. Second para is misleading as BDI is a screening tool rather than diagnostic. Hence the overall prevalence is more of depression symptoms rather than depression.

3. Second para - last 2 sentences were repetitive of result section. This should be discussed, compared or contrast rather than being repeated.

4. "... suggesting that the prevalence of burnout among medical students is fairly similar to that found among medical residents and practicing physicians (27)." reference given was studies on medical students. Hence a proper citation is needed.

5. Last para: The second last sentence was also repeating the result. Some critical discussion is desirable. Why anxiety trait and state could be highly correlated? What could this mean to educators or medical training.

6. Last sentence: Maybe this can help authors to expand the discussion https://www.ncbi.nlm.nih.gov/pmc/articles/PMC6696211/

Strengths and limitations.

1. Need to emphasis the instruments are screening rather than diagnostic.

2. Other limitations have to be discussed - non probability sampling

2. Only captured more of individual constructs. Institutional constructs that may contribute to DABE were not captured

Conclusion only emphasized on high prevalence of DAB. No mention of empathy which was part of the study objective.

6. PLOS authors have the option to publish the peer review history of their article (what does this mean?). If published, this will include your full peer review and any attached files.

Reviewer #1: No

---

## [Author Response · Author response to Decision Letter 0]

21 Oct 2021

Answers marked in red in the Response To Reviewers document.

Reviewer #1: The manuscript offers an overview on medical students mental health in Spain. The strength of the study was the national scale study. However, I feel that the manuscript need to be improved based on the following comments before it is ready for publication.

This manuscript would need a native speaker proofreading to increase the readability of the findings. There is a glaring grammatical and spelling (eg: PubMed) errors throughout the manuscript. We are sorry, the manuscript was revised by a native professional but probably the last additions to the manuscript included some errors. Ms. Diana Draper has revised it again.

Abstract:

Need to include study objective and some general methodology before jumping to results. Worthwhile to include the tools in the abstract so that readers can grasp whether the depression/anxiety prevalence was more of positive symptoms or diagnosis. We agree with the reviewer, and we have included two paragraphs dealing with the objective and the methods employed.

Introduction:

1. The introduction focused only on depression while the constructs measured also include burnout, anxiety and empathy. Hence, authors could incorporate this flow in the introduction. Again, we agree. We have included a new paragraph in the introduction.

Introduction on burnout, depression and anxiety and some overview in general population - High prevalence of burnout, depression and anxiety - why it is more pronounced among medical students - the impact of these mental health problems if not addressed (including empathy) - why empathy is important in medical training - lack of studies looking into this in Spanish context

2. A search in Pubmed (October 6, 2020) revealed 859 articles featuring the Spanish word for "depression" and 517 featuring the Spanish term for “medical students". The union of the two terms returned no records at all.

(From my search, it seems there is such study (PMID: 32920780). Hence, this paragraph should be updated. Yes, the reviewer is right, the article was referenced in our list, but it did not appear in our search then. This spanish journal probably was not indexed immediately.

3. Some introduction on medical training and the structure in Spain is necessary since this is a national scale study. It would help readers to understand which are pre and clinical year students. Ok, we agree and a paragraph has been included in the introduction.

Methods:

1. First para: Need to include population size or estimate to give some picture to readers. Ok, the population number was included originally in the first paragraph of Discussion. This has been also included now in the first paragraph of Methods.

2. Each instruments need to be introduced separately in individual paragraph for a better flow- purpose of the instrument (screening or diagnosis), whether it was administered in English or Spanish version, validation in Spanish or medical students context, with reliability mentioned - scores calculation (whether there is dichotomous classification or measured as continuous variable). Note that validation must be mentioned specifically for the instruments (eg. CBI) rather than the construct (eg. Burnout) We agree, all the instruments ar enow in separate paragraphs.

3. Some explanation is needed to help readers in the difference of anxiety state vs trait. Ok, a new paragraph has been inserted in Methods.

4. What is p75? It is the 75th percentile or third quartile (Q3). This has been included in the revised manuscript.

5. Statistical analysis - more details are needed what are the independent variables included for each logistic regression analyses? How was the independent variables selected for the multivariable model? (Some study used cutoff of p-value more than 0.2 for each univariate analysis run). All the variables (except depression, burnout, anxiety and empathy that were the dependent variables) were the independent variables and we only selected those that gave a significant p value (less than 0.05) in the previous bivariate tests. This has been also added to the paragraph.

Results

1. "Cronbach's alpha coefficients were 0.923 (Beck's test), 0.703 (Maslach), 0.809 (Jefferson), 0.378 (state anxiety) and 0.365 (trait anxiety)." - I think this is not necessary as validation should be done in a separate cohort rather than the actual sample. Yes, we removed this paragraph.

2. Table 1 - Percentage (%) should be in form of 22.9 rather than 22,9. We are sorry, we have corrected it.

3. Table 1 - Age should be excluded and just mentioned in the paragraph in form of mean and standard deviation. Yes, we have included these numbers.

4. Second para: Authors refer this para as Table 1 but it seems there were 6 subgroup analyses that were not in Table 1. These subgroup analyses were mentioned selectively rather than comprehensively. Hence, authors should rethink whether this subgroup analysis need a separate table. The reviewer is right. We added these”subgroups” comments in order to give some more information. Since there are so many tables, we have decided to add the sentence ”data not shown”. We hope you agree, but if necessary we will add these new tables.

5. I suggest that Figure 1 to Figure 5 is changed into Table. It is difficult to read the result precisely from the bar chart. We agree with the reviewer and all these figures have been changed to a table, new table 2. 

6. "Percentage of depression was also significantly higher among 2nd, 3rd, and 4th year students, with those in their 1st and 6th years scoring lowest in this sense (Figure 2)" - How did authors came to this statement. Was one-way ANOVA conducted to ascertain this? - This comment also applies to paragraph on anxiety, burnout and empathy. The term significantly was used only when the chi square p level was lower than 0.05. Then, the rest of sentences just tell the values that are lower and greater, but ANOVA was not performed to obtain significance among them. We have corrected all these sentences to indicate only when there are significant differences. 

7. There was untranslated Spanish words in Table 3. We are sorry, we have corrected it.

8. Drugs and substance abuse: "... with this percentage again being significantly higher among men.." - Authors must be careful when saying significantly unless there was t test or anova done to ascertain the association. Yes, answered in point 6 above.

9. Multivariate relationships between DABE variables and risk predictors - Multivariable instead of multivariate. Ok, we have corrected it.

10. Table 4-7 (Inferior superior should be named as lower and upper). Ok, we have corrected it.

11. p value of 0.000 should be written as p-value < 0.001. Ok, we have corrected it.

Discussion

1. "We therefore believe that the results offer an accurate and highly relevant overview of the mental health of medical students in our country" - I think this is overinflation of the study, plus it is a non-probability sampling. A better way to say this is - the number of participants may reflect the true situation of the problem in Spain medical training. Yes, the reviewer is right. We have changed it in the last lines of the first para of Discussion.

2. Second para is misleading as BDI is a screening tool rather than diagnostic. Hence the overall prevalence is more of depression symptoms rather than depression. Yes, the reviewer is right. We have included the term symptoms in several places in the paragraph.

3. Second para - last 2 sentences were repetitive of result section. This should be discussed, compared or contrast rather than being repeated. The reviewer is right and we have eliminated it.

4. "... suggesting that the prevalence of burnout among medical students is fairly similar to that found among medical residents and practicing physicians (27)." reference given was studies on medical students. Hence a proper citation is needed. Our error, reference should be number 26.

5. Last para: The second last sentence was also repeating the result. Some critical discussion is desirable. Why anxiety trait and state could be highly correlated? What could this mean to educators or medical training. Regarding the correlation between both anxiety state and trait, it is question that still remains unanswered. According to an early formulation, anxiety is a unidimensional construct including both state and trait anxiety, considered to be different sides of the same coin. However, other authors suggested trait and state anxiety to be separate multidimensional constructs.

6. Last sentence: Maybe this can help authors to expand the discussion https://www.ncbi.nlm.nih.gov/ Thank you very much, it is a quite good paper. We have referenced it (new ref. 40) and added a sentence to the last paragraph.

Strengths and limitations.

1. Need to emphasis the instruments are screening rather than diagnostic. Yes, we have added it.

2. Other limitations have to be discussed - non probability sampling. Yes, we also added it.

2. Only captured more of individual constructs. Institutional constructs that may contribute to DABE were not captured. The reviewer is right, we have included a paragraph about it.

Conclusion only emphasized on high prevalence of DAB. No mention of empathy which was part of the study objective. The reviewer is right, we have included a paragraph on empathy.

---

## [Editor Report · Decision Letter 1]

9 Nov 2021

Depression, Anxiety, Burnout and Empathy among Spanish Medical Students

PONE-D-21-11578R1

Dear Dr. Garcia-Estañ,

We’re pleased to inform you that your manuscript has been judged scientifically suitable for publication and will be formally accepted for publication once it meets all outstanding technical requirements.

Kind regards,

Kamran Sattar

Academic Editor

PLOS ONE
---

## [Editor Report · Acceptance letter]

17 Nov 2021

PONE-D-21-11578R1 

Depression, Anxiety, Burnout and Empathy among Spanish Medical Students 

Dear Dr. García-Estañ:

I'm pleased to inform you that your manuscript has been deemed suitable for publication in PLOS ONE. Congratulations! Your manuscript is now with our production department. 

Kind regards, 

on behalf of

Dr. Kamran Sattar 

Academic Editor

PLOS ONE